# Understanding drivers of neonatal mortality in Zimbabwe: A machine learning approach using survey data

Absolom Mbinda[1]*, Rornald Muhumuza Kananura[2], Arsene Brunelle Sandie[2], Richard Makurumidze[1], Agbessi Amouzou[3]

1 Department of Global, Public Health and Family Medicine, Faculty of Medicine and Health Sciences, University of Zimbabwe, Harare, Zimbabwe, 2 African Population and Health Research Center (APHRC), Nairobi, Kenya, 3 Johns Hopkins University Bloomberg School of Public Health, Baltimore, United States of America

* mbindaabso@gmail.com

## Abstract

In Zimbabwe, the neonatal mortality rate (NMR) is higher than the regional average, and the country is not on track to reach the Sustainable Development Goal of reducing the NMR by 2030. While other child mortality indicators have improved, NMR has increased. Using machine learning, we aimed to identify the key predictors of neonatal mortality in Zimbabwe. Pooled secondary data analysis of three rounds of the Zimbabwe Demographic Health Survey (ZDHS) from 2005 to 2015 was done. The study population was all the live births born to women aged 15–49 years within the 5 years prior to each round of the survey (n = 16,941). Multiple supervised binary classification machine learning models were built to predict neonatal death based on socio-economic, mother's demographic, prenatal, delivery, and neonatal characteristics available in ZDHS. Sensitivity and area under the receiver operating curve (AUC ROC) were used to select the best model for the prediction of neonatal mortality. The best model was used to identify relatively important variables, and logistic regression was used to assess the magnitude and direction of effect. The eXtreme Gradient Boosting Model outperformed other models with a sensitivity of 0.74. Early breastfeeding initiation, birth weight, household size, and newborn post-natal care (PNC) were identified as the top predictors of neonatal mortality. Logistic regression revealed that lower birth weight (aOR [95%CI]: 0.9997 [0.9995 – 0.9999]) was positively associated with odds of neonatal mortality, while household size (aOR [95%CI]: 0.84 [0.80 – 0.89]), early breastfeeding initiation, aOR (0.28 [95%CI]: [0.21 – 0.37]) and newborn postnatal care, (aOR [95%CI]: 0.08 [0.06 – 0.11]) were negatively associated with the odds of neonatal mortality. This study demonstrates the potential of machine learning in identifying key predictors of neonatal mortality in Zimbabwe. To accelerate the reduction in neonatal mortality, interventions should focus on preventing and specially managing low birth weight babies to improve survival. Furthermore,

**Data availability statement:** The study was conducted using publicly available data from the Demographic Health Survey Program repository (https://dhsprogram.com/data/dataset_admin/index.cfm). After registering an online account with DHS and submitting an online application stating the research question, access to relevant datasets on DHS repository was granted.

**Funding:** This work was supported, in whole or in part, by the Bill & Melinda Gates Foundation under APHRC's Countdown to 2030's grant number: INV-042414 received by AM. The funder had no role in study design, data collection and analysis, decision to publish, or preparation of the manuscript.

**Competing interests:** The authors have declared that no competing interests exist.

health facilities and community-level support for early initiation of breastfeeding and PNC checks should be promoted for all eligible newborns.

## Introduction

In 2021, approximately 2.3 million children died within the first month of birth, translating to 6,400 neonatal deaths daily. The neonatal period, which is the first 28 days of life, represents the most at-risk period for neonatal and child survival [1]. Neonatal mortality is one of the health and well-being-related indicators for the 2030 Sustainable Development Goals (SDGs) [2]. It is important in measuring the quality of life in the first month and the socio-economic development of a country [3]. Globally, neonatal mortality contributed to 47 percent of under-5 deaths in 2021. The neonatal mortality rate (NMR) has almost halved, decreasing from 37 to 18 neonatal deaths per 1,000 live births between 1990 and 2021 globally. However, variations persist across geographic regions and countries [1,2].

Zimbabwe is one of the countries in the sub-Saharan Africa (SSA) region, with an NMR (32 neonatal deaths per 1,000 live births) that is higher than the regional average (27 neonatal deaths per 1,000 live births) [4]. It is off target on SDG indicator 3.2.2 to reduce NMR to 12 neonatal deaths or fewer per 1,000 live births by 2030 and is only expected to meet the target in 2043 based on current trends [5]. Between 1999 and 2019, NMR increased by 10% in Zimbabwe, while all other under-five and childhood mortalities have been declining according to national surveys. Additionally, its contribution to the under-five mortality rate has increased from about 35% in 2005 to 50% during the same period [2,3,5,6]. Poor quality of services before, during, and soon after pregnancy, as well as poor access to basic emergency maternal and newborn care, have been cited as some of the reasons, as coverage indicators have improved [3,6,7], and the major causes of death have remained the same. Failure to address neonatal mortality will result in the country failing to meet the SDG targets on child and neonatal mortality, sub-optimal quality of life, and violation of basic child human rights to enjoy the highest attainable standard of health. The increase in both the NMR and its contribution to under-5 deaths calls for the need to understand the predictors of neonatal mortality and find innovative ways of identifying newborns at risk in the country, among other things. Various approaches can be used to understand the predictors and identify those at risk, including predictive modeling using machine learning (ML) approaches on existing data.

It is important to understand the predictors of neonatal mortality to help decision-makers identify key policy and program strategies. ML models can be used to identify the predictors of neonatal mortality and their level of importance. This can assist decision-makers at each level of the health system to institute appropriate interventions [8,9]. For instance, with the knowledge of predictors of neonatal deaths, frontline health workers can triage women at risk of having newborn deaths, newborns at risk, and discuss possible outcomes with families and design appropriate selective care pathways.

Global Public
Health

ML methods' use in identifying neonatal mortality factors and predicting neonatal death in developed countries has grown, while in the SSA region, it is still in its infancy, with a few studies done [10,11]. This is despite the differential burden of neonatal mortality, with nearly 98% of the deaths occurring in developing countries [8,12]. Most studies on predictive modeling of neonatal death using ML have been either in high-income countries, focused on special groups such as newborns with certain conditions, or in sub-locations. A neonatal mortality study conducted in 10 countries in SSA using ML and DHS survey data did not include Zimbabwe and used the women of reproductive age (WRA) as the unit of analysis [13]. Our study will focus on Zimbabwe and use each live birth born to WRA as the unit of analysis to account for time variation in antenatal, intrapartum, and postnatal variables' values for children born to the same woman in different years. Other ML studies done in SSA include: Rwanda, on infant mortality [10], Ethiopia and Tanzania, on perinatal mortality [14,15], which looked at a different population from our study. The identification of predictors of neonatal mortality in Zimbabwe has been through traditional statistical methods and only in specific geographic areas [16–21]. To the best of our knowledge, no study has used ML methods and national survey data to identify neonatal factors and attempted to use these factors to build and propose a predictive model to predict neonatal mortality.

ML is gaining prominence in public health [22,23]. The approaches can carry out complex tasks, be applied to large or big data to identify insights and produce reliable predictions, learn and improve performance as new data is added without human intervention, and perform better than traditional statistics [24,25]. Simply stated, ML approaches allow data-driven solutions - that is, they let the data speak for itself [10]. There is a wide selection of ML models or algorithms available to the modeler. These can be used to unmask complex patterns (often non-linear) and interactions and find an optimal solution to the same data [26,27]. Most ML algorithms are flexible, statistical distribution free and do not need specification of assumptions a priori. Some of the popular methods include random forest (RF), K-nearest neighbor (KNN), boosted trees, and logistic regression (LR), among others [9,11]. Using Zimbabwe Demographic and Health Surveys (ZDHS) data collected between 2005 and 2015, we apply these approaches to identify the key predictors of newborn mortality in Zimbabwe.

## Methods

### Study population and data source

Quantitative secondary data analysis of three rounds of the ZDHS 2005/06, 2010/11, and 2015, pooled together, was done to ensure a large enough sample. The data is freely available on the DHS Program [28]. The DHS is an international standard cross-sectional household survey usually conducted every five years to collect data on population, health, and nutrition. In all the rounds, multi-stage cluster sampling was implemented.

The births recode data (BR files), a subset of the individual women's questionnaire, were pooled together for the three survey rounds and used to develop the models to identify the predictors. The key variables of interest were respondents' background characteristics, reproduction, pregnancy, and post-natal care from the women's questionnaire. The unit of analysis was all the live births born to women of reproductive health age (15–49 years) within the 5 years (less than 60 months at the time of the survey) prior to each round of the survey (n = 16,941) for all variables under study. This is the population whose birth history included information on the studied variables.

### Independent and outcome variables

The outcome variable was neonatal death (yes/no) while the independent variables were selected based on literature review [9,11,14,29], the adaptation of the Mosley and Chen [30] framework, and other similarly adapted frameworks by Mfateneza et.al [10] and Titaley et.al [31] on child survival. A neonatal death was counted as a death that occurred within the 1st month (≤ 30 days) of birth in line with how this was defined and calculated in ZDHS, instead of using the exact definition of within 28 days. This was derived from variable B7 (age at death of the child in completed months) in the dataset, which gives a calculated age at death from the reported information. Using variable B7, neonatal death was coded

as 1 = yes = age at death ≤ 1 month, 0 = no = age at death > 1 month, or the child was alive at the time of the interview. The independent variables included were limited to those whose data were collected for all the ZDHS rounds under study and were plausible based on the literature. The considered variables were on socio-economic characteristics of the household and mother, such as household size, antenatal care, delivery, postnatal care, neonatal and environmental characteristics, see S1 Table and S1 Fig in supporting information.

## Model building and data analysis

The modeling process was done in systematic steps, which involved exploratory data analysis, data preparation, and model building, which involved splitting data into train and test, cross-validation, down-sampling, up-sampling, and synthetic minority oversampling (SMOTE) to address class imbalances, and hyperparameter tuning for selected models to avoid overfitting. Lastly, the models were evaluated on test data using metrics derived from the confusion matrix. These steps were derived from standard frameworks used when developing ML models and are explained below concerning this study [26,27]. For reproducibility, we set a random seed for imputation and all ML analyses.

## Sampling

All the unweighted cases (n = 16,941) of data for live births below 60 months (5 years) were used. Survey sampling weights were used to produce frequencies.

## Exploratory data analysis (EDA) and data preparation

Births datasets for the 3 ZDHS survey rounds (2005/6, 2010/11, 2015) were downloaded separately and merged. The combined dataset was filtered by age of the child (< 60 months) at the time of the survey before exploratory data analysis (EDA) was done to identify variable relationships, distributions, options for transformation, data quality issues such as invalid or missing values, and outliers, among other things [26,27]. The births dataset contains data on all births in a woman's lifetime. Births < 60 months of age at the time of the survey were the unit of interest, as their birth history included information on the studied variables. After EDA, the data were transformed, and some variables or their categories were combined. Variables not needed for analysis were dropped to produce a clean dataset for modeling. Variables such as postnatal care signal functions were dropped due to the unavailability of data points in the 2005 and 2011 surveys, respectively. Missing data imputation was done using the missRanger package in R to produce a single dataset (pmm.k = 5, number of trees = 100, case.weights = survey weights). Train and test data were imputed separately, and the outcome variable was included to maintain variable relationships. Data were assumed to be missing at random (see S2 Table for variables that had missing data) after we found associations between missingness in variables in S2 Table and observed variables, and ruled out the missing completely at random assumption. Imputation was done separately for the train and test datasets to avoid data leakage. The clean dataset was one-hot encoded (creating dummy variables for categorical variables) before the models were built. Continuous variables were standardized after the train-test data split to avoid bias toward higher-scaled variables when computing variable importance measures. Correlation of all numerical variables was checked. Birth order and number of children ever born were the only correlated variables. They were all included in the modeling process.

## Modeling

A 70% − 30% stratified random split based on the outcome variable between train and test data was implemented, respectively, before training the models, consistent with other similar studies and commonly used splits [9,11]. The data was split to ensure models were built on training data and evaluated based on test data (data not seen by the model in training) to avoid overfitting. The following widely used [9,11,13] supervised classification models were built, Logistic regression

(LR), Boosted Logistic Regression (LogitBoost), Random Forest (RF), Tree bagging (Treebag), C5.0 decision tree (C5.0), Adapted boosting model (AdaBoost), Support Vector Machine (SVM), General Additive Model (seed), Artificial Neural Networks (ANN), Gradient boosting model (GBM), and Extreme Gradient Boosted (XGBoost) trees. The study outcome was neonatal mortality coded as 1- neonatal death occurrence and 0- not a neonatal death. Given that neonatal mortality is a rare event, it leads to imbalance problems that would bias the prediction results. The distribution of the outcome variable (neonatal death) values was extremely imbalanced, with a ratio of neonatal death to non-neonatal death of 1:32.3. ML methods perform better when the distribution of the values of the outcome variable is closer to 1:1 or 50% and not highly imbalanced. Three methods for balancing data were implemented: down-sampling, up-sampling, and synthetic minority oversampling (SMOTE), and results were compared. The down-sampling procedure within the R caret package balances the data by randomly sampling the dataset in such a way that all classes have equal frequency, with the minority class [15,16]. In contrast, up-sampling within the same package randomly duplicates the minority samples until they have the same class frequency as the majority class. A SMOTE approach was used, where 5 new synthetic minority samples were created utilizing information from existing samples for each minority sample, and the majority cases were down-sampled to achieve equal class frequency.

Implementation of repeated 10-fold cross-validation (cv) repeated 3 times was also done concurrently with resampling procedures (in cv loop) on training data. Hyperparameter tuning using either tune length or a grid appropriate for each ML algorithm was done to optimize results in the inner CV loop. Each model was finally evaluated using test data.

## Feature selection

The Boruta algorithm (a wrapper of the random forest algorithm) was used to select features for the modeling process. Using the training dataset, Boruta started by including all variables under consideration and then statistically confirmed important variables for inclusion and rejected unimportant ones during each iteration round. This was done to remove unimportant variables and reduce the number of features for inclusion in each model [32].

## Model evaluation

Evaluation of the models focused on how accurate or useful the model was in supporting decision-making based on metrics derived from the confusion matrix, Table 1. Accuracy, sensitivity, specificity, and Area under the receiver operating Curve (AUC ROC) were used to select the best model [11,26,27] based on test data. Due to the nature of the problem, correctly classifying neonatal deaths, sensitivity, and AUC ROC were the most prioritized metrics, followed by specificity and accuracy, respectively.

The following formulas show how each of the evaluation metrics is calculated based on the confusion matrix [11,33]. Accuracy, sometimes called percentage correctly classified (PCC) measures the overall proportion of cases correctly classified (Equation 1).

$$Accuracy = \frac{True\ positive\ +\ True\ Negative}{(True\ positive\ +\ True\ Negative\ +\ False\ positive\ +\ False\ negative)} \tag{1}$$

Table 1. Confusion matrix.

| Predicted values | Actual values | |
|---|---|---|
| | **Positive** | **Negative** |
| Positive | True Positive (TP) | False Positive (FP) |
| Negative | False Negative (FN) | True Negative (TN) |

The sensitivity is the true positive rate or the proportion of actual positives classified as positive (Equation 2). In case of neonatal deaths, these will be neonatal deaths correctly classified.

$$Sensitivity\ (Recall) = \frac{\text{True positive}}{(\text{True positive} + \text{False negative})}$$

(2)

Specificity refers to the proportion of actual negative cases correctly classified or true negative rate (Equation 3).

$$Specificity = \frac{\text{True negative}}{(\text{True negative} + \text{False positive})}$$

(3)

### Ethical review

This research utilized the publicly available 2005–6, 2010–11, and 2015 ZDHS data upon receiving permission from the DHS program archivist. No informed consent was required as the researcher used secondary de-identified data and did not interact directly with participants. The researcher complied with all conditions and terms of use of the datasets [34].

## Results

### Descriptive statistics

A total of 17,244 weighted cases for all variables of interest were used for the study. A total of 473 neonatal deaths among the births in the 5 years prior to each survey round were recorded, giving a 10-year neonatal mortality rate (NMR) of 27.4 deaths per 1,000 live births. Generally, from 2005 to 2015, the NMR among the births in the 5 years prior increased by 23% from 23.2 per 1,000 in 2005 to 28.5 deaths per 1000 live births in 2015 (Table 2).

### Descriptive statistics based on identified variables

Most of the neonatal deaths in the study were by mothers from rural areas (69.8%), who had mostly secondary education (60.6%), an average age of [mean ± SE] 19.7 ± 0.05 years, and were not working (54.5%). Nearly half of the neonatal deaths were from poor households (45.1%), from a household with an average (SE) of 5.37 ± 0.25 members, and most (79.3%) of the households used unclean cooking fuel. Most (83.2%) of the neonatal deaths were singletons, with an average birth order (SE) of 2.75 ± 0.14, and they weighed on average (SE) 2,839 grams ± 95.7.

A total of 12% were born within 24 months of the previous birth, and only 17.6% were breastfed within an hour after birth (Table 3).

**Table 2. Neonatal mortality by survey year (n = 17,244).**

| Variables (variable name in database) | Categories | Neonatal Death (N = 473) | Not a Neonatal death (N = 16,771) | Neonatal mortality rate (NMR) *(deaths per 1000 live births)* |
|---|---|---|---|---|
| | | Weighted Frequency n (%) | Weighted Frequency n (%) | |
| Survey round | ZDHS 2005–6 | 121 | 5,110 | 23.2 |
| | ZDHS 2010–11 | 169 | 5,426 | 30.3 |
| | ZDHS 2015 | 183 | 6,235 | 28.5 |
| | **Total** | **473** | **16,771** | **27.4** |

**Table 3.** Socio-economic and demographic, antenatal, neonatal, and environmental characteristics for the births under 5 years in Zimbabwe (combined dataset for DHS 2005-6, 2010-11, and 2015).

| Variables (variable name in database) | Categories | Non-Neonatal deaths Weighted Frequency n (%) | Neonatal deaths Weighted Frequency n (%) | Total Weighted Frequency n (%) |
|---|---|---|---|---|
| **Socio-economic and demographic** | | | | |
| Region (V024) | Manicaland | 2,388 (14.2) | 100 (21.1) | 2,488 (14.4) |
| | Mashonaland Central | 1,771 (10.6) | 46 (9.6) | 1,817 (10.5) |
| | Mashonaland East | 1,492 (8.9) | 34 (7.1) | 1,526 (8.9) |
| | Mashonaland West | 1,990 (11.9) | 77 (16.3) | 2,067 (12.0) |
| | Matabeleland North | 878 (5.2) | 15 (3.2) | 893 (5.18) |
| | Matabeleland South | 747 (4.6) | 7 (1.6) | 754 (4.4) |
| | Midlands | 2,281 (13.6) | 59 (12.6) | 2,340 (13.6) |
| | Masvingo | 2,137 (12.7) | 44 (9.4) | 2,181 (12.7) |
| | Harare | 2,363 (14.1) | 78 (16.4) | 2,441 (14.2) |
| | Bulawayo | 724 (4.3) | 13 (2.7) | 737 (4.3) |
| Type of place of residence (V025) | Urban | 5,066 (30.2) | 139 (29.4) | 5,205 (30.2) |
| | Rural | 11,705 (69.8) | 334 (70.6) | 12,039 (69.8) |
| Education level of mother (V106) | No education | 375 (2.3) | 8 (1.6) | 383 (2.2) |
| | Primary | 5,603 (33.4) | 171 (36.1) | 5,774 (33.5) |
| | Secondary | 10,172 (60.6) | 283 (59.9) | 10,455 (60.6) |
| | Higher | 621 (3.7) | 11(2.4) | 632 (3.67) |
| Religion of mother (V130) | Traditional | 240 (1.4) | 11 (2.0) | 251 (1.5) |
| | Roman catholic, | 1,022 (6.1) | 23 (4.3) | 1,046 (6.1) |
| | Protestant | 2,500 (15.0) | 89 (16.1) | 2,589 (15.0) |
| | Pentecostal | 3,146 (18.8) | 75 (13.7) | 3,221 (18.7) |
| | Apostolic sect | 7,388 (44.3) | 277 (50.4) | 7,665 (44.4) |
| | Other Christian | 1,010 (6.1) | 32 (5.8) | 1,042 (6.0) |
| | Muslim | 77 (0.5) | 0 (0) | 77 (0.4) |
| | None | 1,293 (7.7) | 43 (7.8) | 1,336 (7.8) |
| | Other | 18 (0.1) | 0 (0) | 18 (0.1) |
| Household wealth index quantile (V190a) | Poorest | 3,948 (23.5) | 103 (21.8) | 4,051 (23.5) |
| | Poorer | 3,412 (20.3) | 110 (23.3) | 3,522 (20.4) |
| | Middle | 2,987 (17.8) | 92 (19.3) | 3,079 (17.9) |
| | Rich | 3,674 (21.9) | 112 (23.7) | 3,786 (21.95) |
| | Richest | 2,751 (16.4) | 56 (11.9) | 2,807 (16.3) |
| Household size (v136) | Mean ± SE | 5.51±0.04 | 5.37±0.25 | 5.50±0.04 |
| Number of children ever born (V201) | Mean ± SE | 2.71±0.02 | 3.30±0.14 | 2.72±0.02 |
| Age of mother at first birth (V212) | Mean ± SE | 19.7±0.05 | 19.6±0.34 | 19.7±0.05 |
| Occupation of mother (V717) | Not working | 9,160 (54.6) | 242 (51.0) | 9,402 (54.5) |
| | Professional/managerial bus | 3,998 (23.8) | 96 (20.3) | 4,094 (23.7) |
| | Agricultural | 2,251 (13.4) | 86 (18.3) | 2,337 (13.6) |
| | Household and domestic | 472 (2.8) | 21 (4.4) | 493 (2.9) |
| | Manual | 890 (5.4) | 28 (5.9) | 918 (5.3) |
| **Antenatal care (Pre-Delivery) and intrapartum care** | | | | |
| Antenatal care visits (M14) | No ANC | 1,034 (6.2) | 50 (9.2) | 1,084 (6.3) |
| | 1-3 visits | 2,786 (16.7) | 75 (13.7) | 2,861 (16.6) |

*(Continued)*

| Variables (variable name in database) | Categories | Non-Neonatal deaths Weighted Frequency n (%) | Neonatal deaths Weighted Frequency n (%) | Total Weighted Frequency n (%) |
|---|---|---|---|---|
| | 4 and above | 9,404 (56.3) | 155 (28.2) | 9,559 (55.4) |
| | Missing | 3,471 (20.8) | 269 (50.0) | 3,741 (21.7) |
| Blood pressure taken at least once during ANC (M42C) | No | 885 (5.3) | 18 (39.4) | 903 (5.2) |
| | Yes | 11,460 (68.3) | 172 (36.2) | 11,632 (67.5) |
| | Missing | 4,427 (26.4) | 283 (59.8) | 4,710 (27.3) |
| Iron supplementation during ANC | No | 5,181 (31.0) | 144 (26.2) | 5,325 (30.9) |
| | Yes | 8,038 (48.2) | 136 (24.7) | 8,174 (47.4) |
| | Missing | 3,473 (20.8) | 270 (49.1) | 3,743 (21.7) |
| Ever had a terminated pregnancy (V228) | No | 14,925 (89.0) | 396 (83.6) | 15,320 (88.8) |
| | Yes | 1,809 (10.8) | 77 (16.1) | 1,885 (10.9) |
| | Missing | 38 (2.2) | 1 (0.3) | 39 (0.3) |
| Place of birth (M15) | Home | 4,608 (27.6) | 184 (33.5) | 4,792 (27.8) |
| | Health facility | 11,335 (67.9) | 339 (61.6) | 11,673 (67.7) |
| | Other | 752 (4.5) | 27 (4.9) | 779 (4.5) |
| Skilled attendant at birth (M3A:3B) | No | 4,747 (28.4) 11,948 (71.6) | 194 (35.2) | 4,941 (28.7) |
| | Yes | | 356 (64.8) | 12,304 (71.3) |
| **Neonatal (individual)** | | | | |
| Multiple births (B0) | Single birth | 16,316 (97.3) | 394 (83.2) | 16,710 (96.9) |
| | Multiple | 455 (2.7) | 79 (16.8) | 534 (3.1) |
| Birth interval (B11) | < 24 months (short) | 1,169 (7.0) | 66 (12.0) | 1,235 (7.2) |
| | ≤ 24 - 59 months (normal) | 7,357 (44.1) | 196 (35.6) | 7,553 (43.8) |
| | 60+ months (long) | 3,191 (19.1) | 116 (21.1) | 3,307 (19.2) |
| | First birth | 4,978 (29.8) | 172 (31.3) | 5,150 (29.8) |
| Birth order (BORD) | Mean ± SE | 2.50±0.02 | 2.75±0.14 | 2.51±0.02 |
| Birth weight in grams(M19) | Mean ± SE | 3,130±6.68 | 2,839±95.7 | 3,125±6.93 |
| Birth weight categories (in grams) | Low (<2500) | 1,145 (6.9) | 87 (15.8) | 1,232 (7.1) |
| | Normal (2500 – 3999) | 10,721 (64.2) | 171 (31.1) | 10,892 (63.2) |
| | Large (4000+) | 732 (4.4) | 28 (5.1) | 760 (4.4) |
| | Missing | 4,097 (24.5) | 264 (48.0) | 4,361 (25.3) |
| Early breastfeeding (M34) | No | 7,370 (44.1) | 453 (82.4) | 7,822 (45.4) |
| | Yes | 9,325 (55.9) | 97 (17.6) | 9,422 (54.6) |
| Postnatal care for baby (M70) | No | 2,620 (15.7) | 159 (28.9) | 2,779 (16.1) |
| | Yes | 6,366 (38.1) | 39 (7.2) | 6,405 (37.1) |
| | Missing | 7,709 (46.2) | 352 (64.0) | 8,061 (46.8) |
| **Environmental** | | | | |
| Water source (v113) | Unprotected | 7,419 (44.2) | 226 (47.7) | 7,644 (44.3) |
| | Protected | 9,353 (55.8) | 247 (52.3) | 9,600 (55.7) |
| Type of toilet (v116) | Unimproved | 7,356 (43.9) | 219 (46.4) | 7,576 (43.9) |
| | Improved | 9,415 (56.1) | 254 (53.6) | 9,669 (56.1) |
| Type of cooking fuel (V161) | Unclean | 12,446 (74.2) | 375 (79.3) | 12,821 (74.4) |
| | Clean | 4,326 (25.8) | 98 (20.7) | 4,424 (25.6) |

## Selected variables for model estimation

At the end of the Boruta run, a total of 33 binary and numerical variables were selected, 5 were tentative, and 32 were rejected. Selected variables included wealth index, education, employment status, mother's age at first birth, number of children ever born, place of residence, previous pregnancy termination, ANC attendance, four or more ANC visits, birth interval, birth order, delivery place, early breastfeeding, postnatal care within a month, multiple birth, and birth weight. Water source and clean cooking fuel (Fig 1). This translates to 14.3 events per variable, above the rule of thumb of 10.

The selected variables were used to build the 11 ML models. The caret package in R was used to produce the different evaluation metrics based on the confusion matrix. No variable interactions were considered in advance, as ML methods can identify these patterns automatically. The models were tuned as necessary using tuneLength for all others and tune-Grid for XGBoost.

## Selecting a model for the identification of the best predictors of neonatal mortality

In order of priority, we used sensitivity and AUC ROC as the most important evaluation metrics to compare the models. A model with high sensitivity and a higher AUC ROC can predict neonatal deaths among the neonatal death cases (true positives) in the test data, and at the same time has a good balance between sensitivity and specificity (ability to detect true negatives- non-neonatal deaths).

Sensitivity and AUC ROC results using down-sampling results were overall better than up-sampling and SMOTE sampling for all models. Hence, the selection of the best model was based on down-sampling results. The results show that XGBoost/XGB (74%) performed better than all the other models based on sensitivity, followed by Treebag (72%), Logistic regression (71%), and GAM (71%). When we consider AUC ROC, the boosted logistic regression (LogitBoost) model (81%), GBM (81%), and GAM (81%) performed better than all other models. Precision and F-1 scores for all other models were around 15%, indicating a poor balance between precision and sensitivity, i.e., the models did well in correctly classifying actual neonatal deaths among neonatal deaths but also had high false positives. The values of the evaluation

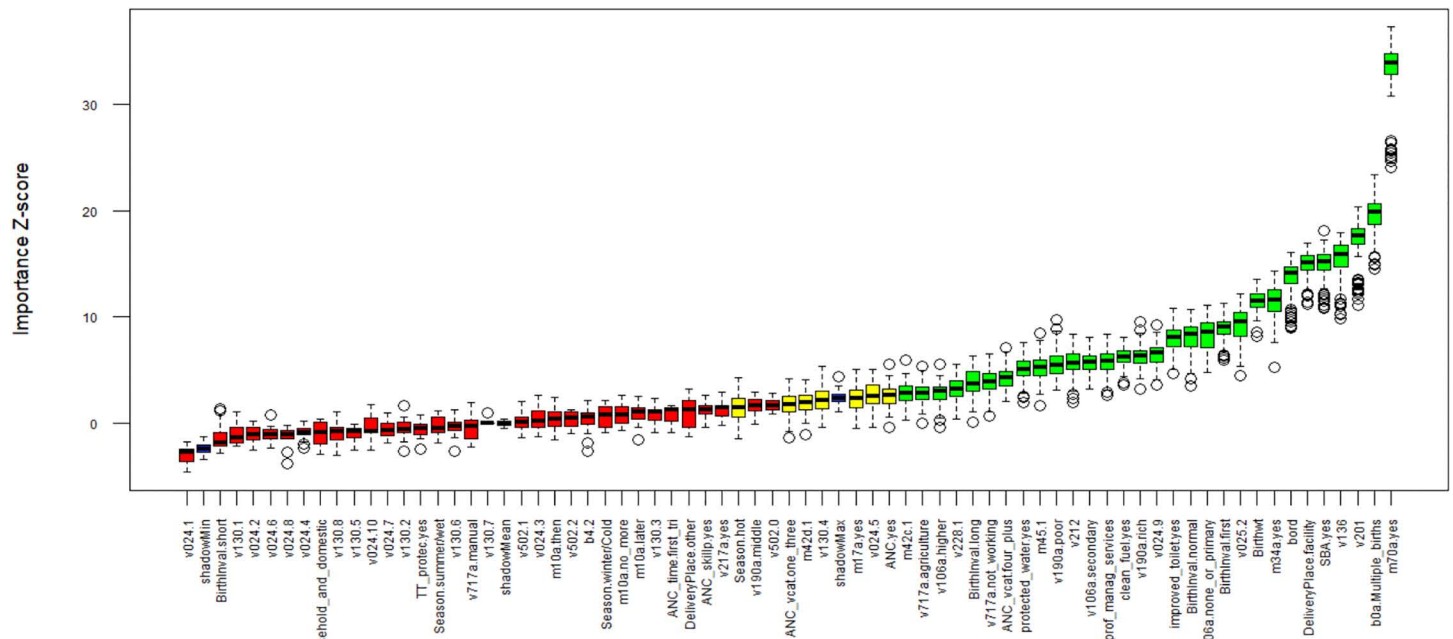

**Fig 1. Variable importance of neonatal mortality predictors based on the Boruta algorithm.**

metrics show that the models built using DHS survey data had average performance. Overall, based on the 2 metrics of sensitivity and ROC, no single model showed predictive superiority over others on all 2 combined, see Table 4.

## Predictors of neonatal mortality using the XGBoost model

A total of 30 out of the 33 variables included in the model-building process were selected by the XGBoost model as relatively important based on the gain metric. Early breastfeeding initiation, birth weight, household size, newborn postnatal care, mother's age at first birth, and number of children ever born were the top six most relatively important variables (Fig 2). The neonatal characteristics (individual) identified as important included early breastfeeding, birth weight, newborn PNC, birth order, multiple births; the mother's predictors included the mother's age at first birth, number of children ever born, occupation, education level (no or primary), iron and folate supplementation during ANC; Household predictors included wealth index, household size, and protected water source among others (Fig 2). Higher education level, birth interval (first birth), and BP check during ANC were not relatively important, with importance scores of zero.

## Assessing the effect direction of the identified neonatal mortality predictors using logistic regression

The top three variables most relatively important in the XGBoost model were also highly significant in traditional logistic regression (early breastfeeding, birth weight, and household size, all with p-values <0.001). However, there were differences; out of the 30 variables selected as relatively important by the XGBoost, only 9 of these variables early breastfeeding, birth weight, household size, newborn PNC, number of children ever born, birth order, multiple births, previous pregnancy termination, and occupation (agriculture) were identified as significant variables in traditional logistics regression. All the variables that were not significant in logistic regression, while relatively important in the XGBoost model, had relative importance scores of less than 10 out of 100, except the age of the mother at first birth (Table 5).

## Household and socio-demographic characteristics of the mother

Household size (adjusted odd ratio (aOR) [95%CI]; 0.84 [0.80 − 0.89]) and occupation of the mother, agricultural (aOR [95%CI]; 1.66 [1.10 − 2.50]), number of children ever born (aOR [95%CI]; 2.91 [2.33 − 3.63]) were the only significant

**Table 4. Model performance evaluation metrics on test data by sampling method.**

| Model | Accuracy (%) | | | Sensitivity (%) | | | Specificity (%) | | | AUC ROC (%) | | |
|---|---|---|---|---|---|---|---|---|---|---|---|---|
| Sampling | Down | Up | Smote | Down | Up | Smote | Down | Up | Smote | Down | Up | Smote |
| RF | 75 | 97 | 75 | 63 | 15 | 39 | 75 | 99 | 75 | 79 | 77 | 66 |
| XGBoost | 69 | 89 | 90 | 74 | 39 | 36 | 69 | 91 | 92 | 78 | 73 | 50 |
| GLM-LR | 74 | 77 | 80 | 71 | 67 | 61 | 74 | 77 | 81 | 77 | 80 | 77 |
| LogitBoost | 77 | 77 | 80 | 68 | 67 | 58 | 77 | 77 | 80 | 81 | 81 | 81 |
| GBM | 78 | 81 | 94 | 69 | 67 | 30 | 79 | 81 | 96 | 81 | 81 | 70 |
| C5.0 | 75 | 97 | 10 | 67 | 15 | 43 | 75 | 99 | 91 | 79 | 79 | 79 |
| Treebag | 72 | 96 | 90 | 72 | 15 | 38 | 72 | 98 | 91 | 79 | 70 | 75 |
| SVM | 74 | 80 | 90 | 69 | 62 | 38 | 74 | 81 | 92 | 79 | 81 | 78 |
| GAM | 75 | 78 | 82 | 71 | 68 | 56 | 75 | 78 | 83 | 81 | 81 | 77 |
| AdaBoost | 74 | 87 | 92 | 67 | 45 | 38 | 75 | 89 | 93 | 80 | 77 | 79 |
| ANN | 73 | 97 | 91 | 65 | 45 | 37 | 75 | 88 | 92 | 77 | 76 | 77 |

Overall, XGBoost was the model with the highest sensitivity, and three of LogitBoost, GBM, and GAM were the models with the highest AUC ROC. The XGBoost had a higher sensitivity, the most prioritized metric between the two, and is better when used to detect more true positive cases, although at the cost of true negatives. Sensitivity was prioritized over AUC ROC to select between the two models, and XGBoost was chosen for subsequent analysis of identifying predictors.

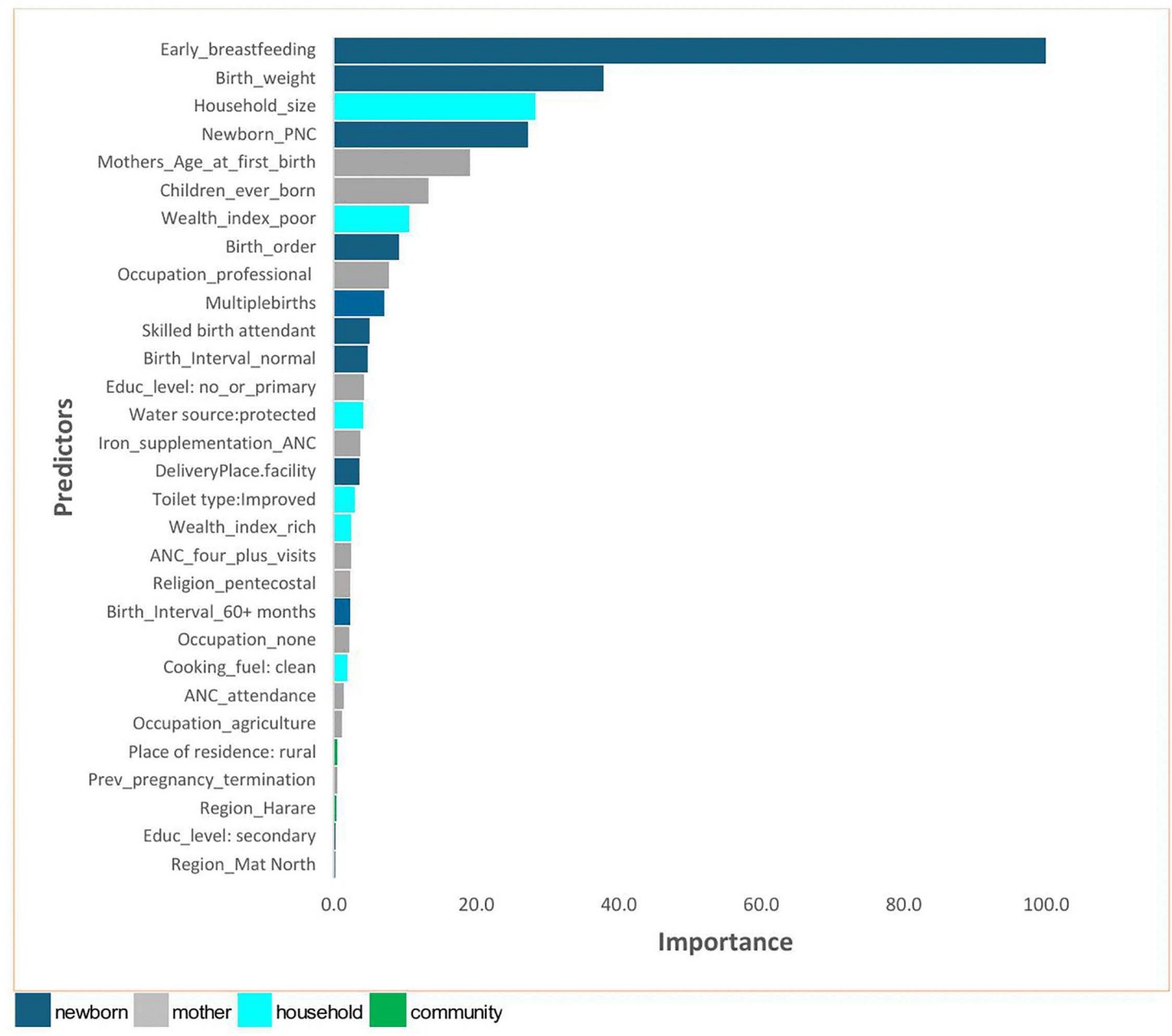

**Fig 2. Important predictors of neonatal mortality based on the XGBoost model.** (Higher values: more important predictors).

variables in logistic regression while in the XGB, region (Harare, Matabeleland North), rural residence, education level (no or primary, secondary), household wealth index (poor and rich), age of mother at first birth, occupation (professional/ management, not working) and religion (pentecostal) were all relatively important (Table 5). None of the household-related environmental variables were significant in logistic regression, while they were all relatively important in XGB.

**Antenatal/pre-natal care.** ANC attendance, four or more ANC visits, iron and folate supplementation during ANC by the mother, and never having had a terminated pregnancy were the relatively important ANC-related variables in the XGB model.

**Table 5. Assessing the effect direction of the identified neonatal mortality predictors using logistic regression adjusted for all variables considered for modeling.**

| Variables (variable name in database) | Categories | Logistic regression | | XGB variable importance |
|---|---|---|---|---|
| | | aOR [95% CI] | (p-value) | |
| **Socio-economic and demographic** | | | | |
| Region (V024) | Matabeleland North | 0.83 [0.45 – 1.47] | 0.533 | 0.2 |
| | Harare | 1.50 [0.91 – 2.47] | 0.111 | 0.3 |
| Type of place of residence (V025) | Rural | 1.21 [0.75 – 1.95] | 0.436 | 0.5 |
| Education level of mother (V106) | Higher | Ref | Ref | |
| | Secondary | 2.13 [0.93 – 5.82] | 0.101 | 0.2 |
| | None or Primary | 2.21 [0.93 – 6.24] | 0.098 | 4.2 |
| Religion | Pentecostal | 1.40 [0.64 –3.39] | | 2.3 |
| Household wealth index quantile(V190a) | Poor | Ref | Ref | 10.6 |
| | Middle | 1.08 [0.75 – 1.54] | 0.672 | - |
| | Rich | 1.42 [0.88 – 2.28] | 0.147 | 2.4 |
| **Household size (v136)** | | 0.84 [0.80 – 0.89] | <0.001* | 28.3 |
| **Number of children ever born (V201)** | | 2.91 [2.33 – 3.63] | <0.001* | 13.3 |
| **Age of mother at first birth (V212)** | | 1.01 [0.97 – 1.05] | 0.713 | 19.1 |
| **Occupation of mother (V717)** | Professional/mangmnt | Ref | Ref | 7.7 |
| | Domestic/Manual | 1.24 [0.77 – 1.97] | 0.358 | - |
| | **Agricultural** | 1.66 [1.10 – 2.50] | 0.016* | 2.2 |
| | Not working | 0.95 [0.69 –1.33] | 0.768 | 1.1 |
| **Environmental** | | | | |
| Water source (v113) | Protected | 1.20 [0.92 – 1.57] | 0.176 | 4.1 |
| Type of toilet (v116) | Improved | 0.73 [0.52 – 1.01] | 0.058 | 2.9 |
| Type of cooking fuel (V161) | Clean | 0.92 [0.59 – 1.42] | 0.696 | 1.9 |
| Antenatal care, neonatal, intrapartum, and post-natal care | | | | |
| ANC attendance | Yes | 0.98 [0.55 – 1.77] | 0.943 | 1.4 |
| Four or more ANC visits | Yes | 0.92 [0.70 – 1.23] | 0.583 | 2.4 |
| Iron supplementation (M45) | Yes | 1.02 [0.76 – 1.37] | 0.912 | 3.6 |
| **Ever had a terminated pregnancy (V228)** | Yes | 1.46 [1.04 – 2.02] | 0.025* | 0.5 |
| Place of birth (M15) | Health facility | 1.54 [0.83 – 2.93] | 0.181 | 3.6 |
| Skilled birth attendant | Yes | 1.59 [0.86 – 2.90] | 0.133 | 5 |
| **Multiple births (B0)** | Yes | 2.89 [1.92 – 4.30] | <0.001* | 7 |
| Birth interval (B11) in months | ≤ 24 - 59 (normal) | Ref | Ref | 4.8 |
| | 60+ (long) | 1.32 [0.94 -1.83] | 0.102 | 2.2 |
| **Birth order (BORD)** | | 0.41 [0.33 – 0.52] | <0.001* | 9.2 |
| **Birth weight** | | 0.9997 [0.9995 – 0.9999] | <0.001* | 37.9 |
| **Early breastfeeding (M34)** | Yes | 0.28 [0.21 -0.37] | <0.001* | 100 |
| **Postnatal care for baby (M70)** | Yes | 0.08 [0.06 – 0.11] | <0.001* | 27.3 |

*Significant variables in traditional logistic regression.

All these were not statistically significant in logistic regression, except those who had ever had a terminated pregnancy, who had a 46% increase in odds of having a neonatal death as compared to those who had never terminated a pregnancy (Table 5).

**Intrapartum/delivery and post-natal care.** Early initiation of breastfeeding and postnatal care attendance for the newborn were statistically significant in the logistic regression and were among the top four relatively important predictors

of neonatal death in the XGB model. Early breastfeeding (within an hour of birth) was associated with a 78% reduction in odds of neonatal death as compared to those who did not breastfeed within an hour (aOR [95%CI]; 0.28 [0.21 -0.37]). Receiving newborn PNC was associated with a 92% reduction in odds of neonatal mortality as compared to those who did not receive PNC (aOR [95%CI]; 0.08 [0.06 – 0.11]). Delivery at the facility and getting a skilled birth attendant were identified as relatively important in the XGB model, but were not statistically significant in the logistic regression (Table 5).

**Neonatal predictors.** Three variables, multiple births (aOR [95%CI]; 2.89 [1.92 – 4.30]), birth order (aOR [95%CI]; 0.41 [0.33 – 0.52]), and birth weight (aOR [95%CI]; 0.9997 [0.9995 – 0.9999]) were significant in logistic regression. Multiple births had 2.89 times higher odds of death compared to singleton births. An increase in birth order was associated with decreased odds of neonatal death, while an increase in birth weight by 100g was associated with a 3% reduction in odds of neonatal death. In XGB, 4 variables were relatively important: birth interval, and the 3 variables that were significant in logistic regression (Table 5).

## Discussion

To the best of our knowledge, this study is the first in Zimbabwe that has used ML approaches to develop prediction models and identify neonatal mortality predictors using national survey data. This study contributes to knowledge in public health research on the identification of neonatal mortality predictors and predictive modeling using ML techniques and traditional methods to understand the effect direction. The ML approach identified predictors that could be grouped as community, household, mother's characteristics, and neonatal, antenatal, intrapartum, and post-natal characteristics. The top three relatively important variables, as identified using the ML approach, were all statistically significant when analyzed using the traditional logistic regression. Additional variables were identified, although they were not significant in logistic regression, and this could be due to ML's ability to identify non-linear effects. This implies that ML is a feasible method to identify predictors in public health and can complement or substitute traditional/ classical statistical methods in some circumstances. The advantages of ML over traditional statistical methods have been well documented [25,35], and this could be the reason for the observed results, as ML methods can uncover complex relationships that sometimes can be missed by traditional methods. The discussion will focus on explaining the top ten predictors.

### Individual neonatal predictors

Birth weight, birth order, multiple births, and birth interval were the three neonatal predictors identified in this study as relatively important in that order. The sex of the child was not found to be relatively important, although some studies found it to be important [11,36–38]. Birth weight was the second most relatively important predictor, and this is in line with other studies that have highlighted birth weight as one of the most important predictors of neonatal mortality [9,19,39,40] if not the single most important one [12]. Birth weight is a function of overall pre-conception maternal nutrition and health status, and care given during pregnancy. Some studies have shown that low birth weight is associated with an increased risk of fetal or neonatal mortality [19,41]. This suggests that public health programs should focus on both preventive (ensuring adequate nutrition for pregnant women, treating illnesses, providing quality antenatal care) and curative (e.g., thermal care, feeding of the baby) components [42]. Low birth weight is common in developing countries, which is also an indicator of inequality among regions and income groups [43,44]. Addressing these inequalities has the potential to reduce the incidence of low birth weight and improve neonatal outcomes [45]. High-impact interventions such as immediate kangaroo care, provision of antenatal corticosteroids, and tocolytic treatment, among other interventions to address low birth weight/ prematurity, need to be scaled up as well as ensure they are implemented with fidelity to realize better outcomes [42,46]. Additionally, to have an impact, models for risk identification should consider birth weight, birth order, multiple births, and birth interval as important individual neonatal predictors of mortality.

Similar to other studies, birth order was found to be a relatively important factor [41]. The first baby and the 4th and above babies were found to be at increased risk [25,47,48]. Competition for resources by siblings and increased demand

biologically on the mother to care for more babies may explain the risk of higher-order babies [47,49], while for the first baby, the mother may still be a teenager at the time of birth [48]. This implies that special care should be taken, and these babies should be born in properly equipped facilities. During antenatal and prenatal care, mothers should also be made aware of the risk associated with higher birth order through health promotion.

Multiple births were found to be associated with higher odds of death as compared to singleton births, which is in line with other studies [11,36,38]. Multiple births are likely to be born with low birth weight or premature, which may increase their risk of death. The mothers are also likely to experience pregnancy or delivery complications, which affect the outcome. Additionally, fetal growth restriction can also contribute to mortality [38]. This suggests that there is a need for close monitoring of multiple pregnancies and births by skilled personnel before, during, and after birth to realize better outcomes.

### Mother's characteristics associated with neonatal mortality

Mother's age at first birth, number of children ever born (parity), no occupation, professional occupation, education level (secondary, none, or primary), Pentecostal religion, and agricultural occupation were the socio-economic characteristics of the mother that were relatively important, and this is in line with other studies [29]. Mother's age at first birth, number of children ever born, and occupation (professional) were among the top ten relatively important variables, while the other variables' influence was relatively low. Mothers who start childbearing below 18 years and those who start late, at 40 years or above, are at higher risk [19], and the education of women on pregnancy planning with respect to their age may help improve outcomes.

The type of occupation is potentially correlated with income in developing countries, and implementing livelihood programs is one way to address this challenge. Similarly, with other studies, the number of children ever born was found to be relatively important. An increase in the number of children born was associated with increased risk of neonatal death [25,29,36]. Interventions to promote contraception use for women of reproductive health to space and limit births should be encouraged [36]. While this study found some socio-economic variables of the mother to be important, this is contrary to another study done locally at 2 major hospitals, which found that no socio-economic variable was associated with neonatal mortality [19]. The geographic scope of the survey may have resulted in different results.

### Household socio-economic and environmental predictors

Household size and wealth index were the household socio-economic predictors, while toilet type, water source, and cooking fuel were the environmental predictors identified in this study. While the other variables' relative influence was not that large, household size was the third most relatively important predictor in this study, after early breastfeeding and birth weight, and was negatively associated with the odds of neonatal mortality. This finding is unanticipated, in sharp contrast to most studies, which reported household size to be positively related to odds of neonatal mortality, but corroborates an unexpected finding found in a similar ML study done in SSA [13] and another study done in Ghana [36]. This finding may suggest that large households have readily available and better support systems than small households, but this needs to be further investigated empirically [13]. Wealth index, a measure of inequality, was also found to be among the top ten predictors of neonatal mortality. This corroborates with other studies [29] and underscores the need to address disparities to achieve good neonatal outcomes.

### Pre-natal, delivery, and post-natal care predictors

ANC attendance, four or more ANC visits, iron and folate supplementation during prenatal care, and previous pregnancy termination were the identified prenatal predictors in line with other studies [13,19,50], although their influence in the model was relatively low. Health facility delivery and delivery by a skilled birth attendant were the delivery-related

predictors, and this is in line with other studies [38]. Early breastfeeding initiation was the top relatively important predictor, while newborn postnatal care was among the top four factors associated with neonatal mortality. Contrary to other studies, delivery by c-section was not found to be a predictor [13,19].

Receiving newborn postnatal care was found to be protective in this study, and this underscores the need for mothers and their babies to access and utilize appropriate health services soon after delivery [31]. Attending postnatal care increases the likelihood of getting the various interventions as needed that help screen for danger signs and reduce mortality. Mothers and family members should be made aware of the importance of postnatal care checks to ensure that even those who deliver at home are also checked, and community follow-ups can be encouraged during the first month, with a special focus on those identified to be at risk. The quality of the postnatal care services should also be enhanced to realize better outcomes.

Early breastfeeding initiation was the top relatively important predictor associated with neonatal mortality in our study and it was highly protective. This is in line with other studies, which found it to be important [51]. Initiation of breastfeeding within 1 hour of birth is one of the WHO-recommended easy and cost-effective interventions to reduce neonatal deaths and improve survival and nutrition even beyond the neonatal period [51,52]. Early breastfeeding is important to reduce neonatal infections (e.g., sepsis, diarrhea, pneumonia) and prematurity-related deaths and establish exclusive breastfeeding in the long term and get the associated benefits [53]. According to other studies, it can reduce neonatal deaths by 20% and has a long-lasting impact on survival. Delaying breastfeeding by an hour was noted to increase neonatal mortality by 2-fold in other studies [52]. Colostrum, which is the first milk, is rich in nutrients that protect against these infections and other diseases. Cultural and religious practices, awareness, individual socioeconomic factors, and opinions from in-laws have been noted to influence initiation of breastfeeding and uptake of colostrum in other studies [53], but these may need to be explored in our own context, as they were noted to vary by region in other studies. In our study, the prevalence of early breastfeeding was low (55%). Awareness of its importance through various channels, such as health provider education and support at health facilities, the community through peer support and volunteers [54], and mass media, can help promote early initiation. Furthermore, a breastfeeding policy can be established and implemented at health facilities and communities. Additionally, other interventions such as the implementation of skin-to-skin contact and baby Baby-Friendly Hospital Initiative at scale and with fidelity may yield positive results.

## Study strengths and limitations

This study used ZDHS data for 3 rounds, which spanned over 10 years. The ZDHS data is national data. This data was collected for other purposes and not solely for this study, and some key variables potentially associated with neonatal mortality were missing in the dataset or were dropped because they were not collected in all rounds of ZDHS. These variables include clinical and quality of care variables such as gestation age at birth, resuscitation, Apgar score, thermal care, abnormalities, and post-natal signal functions performed among others. Despite this, our study was strong in showing some key socio-economic and demographic predictors while also controlling for the few basic prenatal and delivery variables that were available.

The DHS is a cross-sectional survey with self-reporting and prone to recall bias on some health access variables, although this is expected to be less for mortality-related variables, and the DHS tries to mitigate this by reducing the recall period to 2 years for prenatal and postnatal care questions. A cross-sectional survey indicates variables that are associated with the outcome and does not infer causality. Pooling datasets may have introduced temporal bias; we controlled for survey year in our analysis. Imputation of missing variables may have introduced some bias in the results, although this is expected to be low compared to the complete cases analysis. Minimal survivor bias is expected for PNC and early breastfeeding initiation variables, and these should be interpreted cautiously. We compared results from different data balancing methods and understand how results differ.

ML algorithms, including the XGBoost model, identify a set of variables that are relatively important in a model and give the highest predictions when combined, but do not give the direction of the association between outcome and

independent variable [33]. In addition, some variables may be associated with the outcome without necessarily being selected in the set of important variables that yield high predictions. In the real world, predictors work in synergy, and ML models used in this study were good at identifying the set of predictors that best predict neonatal mortality. This may be important in building risk identification models and selecting the most impactful interventions in public health based on a set of key predictors identified. To aid in understanding the magnitude and direction of association for the relatively important variables identified through ML, traditional logistic regression was built. The XGBoost model is a gradient boosting ensemble method that has advanced regularization options to prevent overfitting. It improves its accuracy based on learning from previous model iteration errors, handles high-dimensional and large datasets well, is computationally efficient, and scalable. Its major drawback is that it requires careful selection and tuning of hyperparameters. It's also complex but interpretable [55]. External validation and model calibration were out of the scope of this study. While sensitivity was high, the F1-scores were low, implying high false positive rates. This is not a big concern compared to false negatives.

## Conclusion

The descriptive analysis showed that the neonatal mortality rate in Zimbabwe has increased over the 10-year period of the study. Using ML methods, we identified a set of key factors associated with neonatal mortality in Zimbabwe, some of which could not be identified previously using traditional statistical approaches. This indicates that ML is a feasible method of identifying factors of neonatal mortality and can complement traditional statistical methods in many ways.

If Zimbabwe is to accelerate progress on reducing the neonatal mortality rate, policymakers and programmers should look at addressing some of the key predictors identified in this study. The top 10 key predictors of neonatal mortality identified in this study during the period were early breastfeeding, birth weight, household size, newborn PNC, age of mother at first birth, number of children ever born, wealth index-poor, birth order, occupation, and multiple births, respectively. Some of these key predictors and socio-economic predictors identified in this study, such as wealth index, place of residence, and educational level, are measures of deprived individuals and communities that can be linked to health inequalities. We argue that a holistic multi-sectoral approach that also focuses on addressing health inequalities and socio-economic disparities at large will help reduce neonatal mortality instead of heavily investing in curative components while attention to other components lags. Community and health facility awareness and promotion of early initiation of breastfeeding in the immediate postpartum period is an easily attainable and low-cost intervention that has been proven in other settings and can bear long-term benefits. A policy to support early breastfeeding can be established, or the implementation of the same strengthened where it exists. Preventive interventions to address issues of low birth weight, the second most important predictor, need to be considered and can start from addressing nutrition issues before conception and during pregnancy, as well as screening for illnesses of mothers during prenatal care to reduce the incidence of low birth weight. High-impact low-cost interventions to address low birth weight/prematurity, such as care for small and sick newborns, can also be reinforced and scaled up [42]. Strengthening of quality of postnatal care services, starting with key interventions that are cost-effective, such as breastfeeding counseling and assessing danger signs, is key.

The key predictors identified in this study can be used to identify mothers and newborns at risk and design appropriate intervention packages that address the preventive and curative components of maternal and newborn programs. In this study, we used survey data that had some limitations and ended by identifying predictors. For future studies, we recommend the use of routinely collected patient-level data adjusted to collect all necessary variables and deploy the model for use to utilize the capabilities of ML algorithms as electronic decision support tools in health, and identify mothers or newborns at risk of death, offer selective care pathways, and improve outcomes of mothers and newborns. Additionally, it would be valuable to explore the specific causes of the risk factors identified and investigate causal relationships or interactions between these variables to gain an understanding of the underlying dynamics and predictors of neonatal mortality.

## Supporting information

**S1 Fig. Conceptual framework for socio-economic, antenatal, intrapartum, and postnatal care factors.**
(DOCX)

**S1 Table. Socio-economic, demographic, antenatal, intrapartum, and postnatal care variable types, definitions, and categories.**
(DOCX)

**S2 Table. Variables with missing values (N = 16,941).**
(DOCX)

## Acknowledgments

The authors would like to thank the following for their support during the study: the Countdown 2030 Cohort 1 fellowship team of supervisors, trainees, and the African Population and Health Research Center fellowship coordination team. This manuscript is a result of work that was originally conceptualized by AM as part of the Master's in Data Science project at the University of East London.

## Author contributions

**Conceptualization:** Absolom Mbinda.

**Data curation:** Absolom Mbinda, Rornald Muhumuza Kananura.

**Formal analysis:** Absolom Mbinda, Rornald Muhumuza Kananura.

**Methodology:** Absolom Mbinda.

**Supervision:** Rornald Muhumuza Kananura, Arsene Brunelle Sandie, Richard Makurumidze, Agbessi Amouzou.

**Visualization:** Absolom Mbinda, Rornald Muhumuza Kananura.

**Writing – original draft:** Absolom Mbinda.

**Writing – review & editing:** Absolom Mbinda, Rornald Muhumuza Kananura, Arsene Brunelle Sandie, Richard Makurumidze, Agbessi Amouzou.

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
