## [Decision Letter · Decision Letter 0]

21 Jan 2025

PGPH-D-24-02248

Understanding drivers of neonatal mortality in Zimbabwe: A machine learning approach using survey data.

Dear Dr. Mbinda,

Thank you for submitting your manuscript to PLOS Global Public Health. After careful consideration, we feel that it has merit but does not fully meet PLOS Global Public Health’s publication criteria as it currently stands. Therefore, we invite you to submit a revised version of the manuscript that addresses the points raised during the review process.

We look forward to receiving your revised manuscript.

Kind regards,

Mufaro Kanyangarara

Academic Editor

Journal Requirements:

1. Please note that PLOS Global Public Health has specific guidelines on code sharing for submissions in which author-generated code underpins the findings in the manuscript. In these cases, all author-generated code must be made available without restrictions upon publication of the work. Please review our guidelines at https://journals.plos.org/plosglobalpublichealth/s/materials-and-software-sharing#loc-sharing-code and ensure that your code is shared in a way that follows best practice and facilitates reproducibility and reuse.

2.  Please provide an Author Summary. This should appear in your manuscript between the Abstract (if applicable) and the Introduction, and should be 150–200 words long. The aim should be to make your findings accessible to a wide audience that includes both scientists and non-scientists. Sample summaries can be found on our website under Submission Guidelines:

https://journals.plos.org/globalpublichealth/s/submission-guidelines#loc-parts-of-a-submission.

Additional Editor Comments (if provided):

Reviewers' comments:

Reviewer's Responses to Questions

**Comments to the Author**

1. Does this manuscript meet PLOS Global Public Health’s publication criteria?

Reviewer #1: Partly

Reviewer #2: Yes

Reviewer #3: Yes

Reviewer #4: Yes

2. Has the statistical analysis been performed appropriately and rigorously?

Reviewer #1: Yes

Reviewer #2: Yes

Reviewer #3: Yes

Reviewer #4: Yes

3. Have the authors made all data underlying the findings in their manuscript fully available (please refer to the Data Availability Statement at the start of the manuscript PDF file)?

Reviewer #1: Yes

Reviewer #2: Yes

Reviewer #3: No

Reviewer #4: Yes

4. Is the manuscript presented in an intelligible fashion and written in standard English?

Reviewer #1: No

Reviewer #2: Yes

Reviewer #3: Yes

Reviewer #4: Yes

Reviewer #1: Zimbabwe’s neonatal mortality rate (NMR) remains high, off track to meet the 2030 SDG targets, and unlike other child mortality indicators that have improved, NMR has remained stagnant. This study uses machine learning to identify key predictors contributing to neonatal mortality in Zimbabwe, analyzing pooled data from the 2005-2015 Zimbabwe Demographic and Health Survey (ZDHS). Based on sensitivity, and Area Under the Receiver Operating Curve (AUC ROC), Gradient Boosting Model (GBM) was selected as the top-performing model for predicting neonatal mortality. Using this model Birth weight, household size, and birth order were identified as significant predictors while logistic regression showed that lower birth weight and higher birth order increased neonatal mortality risk, while larger household size had an inverse relationship.

This study offers valuable insights into the predictors of neonatal mortality, holding particular significance as the first study in Zimbabwe to utilize machine learning for identifying these factors. However, to enhance clarity and readability, the manuscript would benefit from grammar revision and edits. This includes breaking up lengthy paragraphs, ensuring correct tense usage, refining phrasing, and incorporating appropriate punctuation. Additionally, below are my specific feedback on each section to consider for further refinement.

The introduction effectively presents the global and regional context of neonatal mortality, narrowing down to Zimbabwe’s specific challenges in Sub-Saharan Africa. The authors clearly highlight the role of ML and need (identified gap) for this study.

The Methods is well-organized and clearly explains the model selection.

The result section is well crated. Tables and figures were used when appropriate. The authors also have done an excellent job of comparing outputs from both linear regression and machine learning models. This approach adds depth to the analysis and helps understand similarities and distinction between these methods. and demonstrating how they complement each other.

The Discussion section provides valuable insights and compares the results well with existing literature. However, some of the statements should be strengthened by incorporating proper references (citations) to enhance the credibility and provide a clearer basis for drawing conclusions based on the findings.

Strengths and Limitations section effectively highlights the study's strong points and acknowledges its constraints particularly with respect to the data set and machine learning approach. Adding limitations with using the specific GBM model applied would make it more contextual for interpreting the findings.

The conclusion effectively summarizes the role of machine learning in identifying predictors for neonatal mortality. However, I would suggest revising phrases like "I argue" in favor of a unified authorial voice (using “we”) ensuring the recommendations reflect a team consensus. In cases where consensus has not been achieved, this should be indicated. Including references to support proposed argument/strategies would also enhance the credibility and feasibility of the recommendations.

Additionally, for future studies, it would be valuable to explore the specific causes of the risk factors identified and investigate causal relationships or interactions between variables. This could yield more comprehensive insights into the underlying dynamics driving neonatal mortality, allowing for the design of more targeted and effective interventions.

Overall, this manuscript offers timely and valuable insights into the predictors of neonatal mortality using machine learning in the country. The findings provide a strong foundation for future research and could inform targeted interventions to reduce neonatal mortality rates (NMR). However, the manuscript would benefit from grammar revisions and a more precise use of citations/references to align with publication standards. Incorporating this and the above-mentioned suggestions would further enhance clarity, and overall quality of the manuscript.

Reviewer #2: Thank you for the opportunity to review this article

The manuscript is generally well-written. However, I have a few concerns that should be addressed before publication.

Introduction section

I recommend modifying lines 50 to 52 as follows, “Neonatal mortality rate (NMR) has almost halved, decreasing from 37 to 18 neonatal deaths per 1,000 live births between 1990 and 2021 globally. However, variations persist across geographic regions and countries (1,2).” Instead of “Neonatal mortality rate (NMR) has almost halved from 37 to 18 neonatal deaths per 1,000 live births between 1990 and 2021 globally but there remain variations across geographic regions and countries (1,2).”

Line 53 and 54 should be modified to make the sentence clear and easy to comprehend.

Results

Line 306 should be written as “Two variables, birth order (aOR [95%CI]; 1.29 [1.14 - 1.45]) and birth weight (aOR [95%CI]; 0.99 [ 0.998 – 0.999]) were significant in logistic regression.” Instead of “Two variables, birth order (aOR [95%CI]; 1.29 [1.14 - 1.45]), birth weight (aOR [95%CI] and 0.99 [ 0.998 – 0.999]) were significant in logistic regression.”

Discussion

From lines 357 to 361, why were the reasons for the birth order result not stated/explained in the discussion, unlike those for birth weight and birth interval? I recommend doing so.

Why was ‘birth interval’ discussed as a predictor in the discussion but was not included in the conclusion?

Answers

The above grammatical errors should be corrected to ensure clarity and easy comprehension of the article

Reasons should be stated for each findings in the study with references from other articles to support claim. Findings should not just be stated in the study.

If birth interval was found to be a predictor, it should be included in the conclusion after discussion.

Congratulations to the entire team for this study.

Reviewer #3: Abstract: It would be better if you write a structured abstract that includes introduction, objective, material & method, Results and conclusion, in that order.

Datasharing: Publish the data related to your analysis

NB: Generally, a very good study

Reviewer #4: Introduction

Lines 57, 57 “NMR has stagnated in Zimbabwe over the past two decades while all other under 5 and childhood mortalities have been declining.”

This information needs to be verified further using local data sources such as DHIS. NMR appear to be increasing in Zimbabwe since 2019.

Method

Line 107. “The outcome variable was neonatal death (yes/no)”

How did you arrive at yes/no (mortality or otherwise) from the data sets?

Line 108. “While the independent variables were selected based on adaptation of the Mosley and Chen framework and other similarly adapted frameworks…”

It is probably clearer to list/name adapted sources rather than just providing reference.

Lines 121 to 123. “The modeling process was done in systematic steps which involved 120 exploratory data analysis, data preparation, model building which involved splitting data into train and test, cross validation, down sampling to address imbalances, hyperparameter tuning for selected models to avoid overfitting”

Entire paragraph sounds theoretical. Authors should perhaps describe how you arrived at the model used without re-stating theoretical phrases.

Line 131. “Merged births dataset was filtered by age of child (< 60 months) at time of survey before exploratory data analysis (EDA) was done”

What was the rationale for a merger and why was there the need to filter? DHS data across the rounds are supposed to be identical. This is not clear to me.

Outcome variable

Although the authors stated that neonatal mortality was the outcome of interest as binary, this variable does no appear anywhere in the model/Table. You main variable need to come clear and reported in the Table.

Results

Line 36 “while household size had an inverse relationship with neonatal mortality (aOR [95%CI]: 0.87 [0.79 – 0.95]).”

1. Do you mean with increased household size, there is a corresponding decrease in neonatal mortality? In that case why is your recommendation advocating for the use of family planning to regulate household size?

Table 5. “Household size (v136) 0.87 [0.79 – 0.95] <0.001*”

Is this variable “household size” continuous variable? In any case the interpretation of the results could is problematic despite the statistically significant p-value. You are suggesting that increased household size led to decrease neonatal mortality, but your outcome variable is binary (yes/no) and this is tricky because your finding suggest that an increase of 1 unit in family size lead to a decrease of 13% neonatal mortality. Your outcome of interest is binary (yes/no) dead or alive which is why you chose logistics regression. I don’t see how you can explain the influence of household size in the manner that you did i.e. inversely related. You probably need to drop the household variable or categorize it to allow for a meaningful interpretation to avoid misleading readers.

Table 5. “Age of mother at first birth (V212) 0.99 [ 0.94 – 1.06] 0.950 29.45”

What are your criteria for determining predictors and why does “Age of mother” matter as capture in concluding discussion? Age is not categorized, and not statistically significant. How did you arrive at its relevance as a predictor?

Discussion

Line 340. “Birth weight is a function of maternal nutrition, health status and care during pregnancy”

I’m not sure if the maternal nutrition, and health status are mutually exclusive from care during pregnancy. Outcome of functions over lapses. Sentence should be re-phrased.

Overall

Useful study that would contribute to literature and drive policy development in support of the fight against neonatal mortality in Zimbabwe.

**Do you want your identity to be public for this peer review?** For information about this choice, including consent withdrawal, please see our Privacy Policy

Reviewer #1: No

Reviewer #2: **Yes:** Edmund Mintah Wiafe

Reviewer #3: No

Reviewer #4: **Yes:** Dr John Azaare

---

## [Decision Letter · Decision Letter 1]

19 May 2025

PGPH-D-24-02248R1

Understanding drivers of neonatal mortality in Zimbabwe: A machine learning approach using survey data.

Dear Dr. Mbinda,

Thank you for submitting your manuscript to PLOS Global Public Health. After careful consideration, we feel that it has merit but does not fully meet PLOS Global Public Health’s publication criteria as it currently stands. Therefore, we invite you to submit a revised version of the manuscript that addresses the points raised during the review process.

Two reviewers have suggested a reject decision at this stage due to concerns with the machine learning component of your study. Please review their comments and make the necessary amendments to your manuscript and provide point by point responses. In particular, one reviewer notes the lack of model validation. It is a requirement of publication in PLOS Global Public Health that experiments, statistics, and other analyses are performed to a high technical standard and are described in sufficient detail (https://journals.plos.org/globalpublichealth/s/criteria-for-publication#loc-3). Specifically, where your work uses machine learning techniques, it must meet the criteria of utility, validation, and availability, which are described in detail at https://journals.plos.org/globalpublichealth/s/submission-guidelines#loc-methods-software-databases-and-tools.

We look forward to receiving your revised manuscript.

Kind regards,

Joanna Tindall, PhD

Staff Editor

Additional Editor Comments (if provided):

Reviewers' comments:

Reviewer's Responses to Questions

**Comments to the Author**

Reviewer #1: All comments have been addressed

Reviewer #2: All comments have been addressed

Reviewer #3: All comments have been addressed

Reviewer #4: All comments have been addressed

Reviewer #5: (No Response)

Reviewer #6: All comments have been addressed

publication criteria?

Reviewer #1: Yes

Reviewer #2: Yes

Reviewer #3: Yes

Reviewer #4: Yes

Reviewer #5: Partly

Reviewer #6: No

3. Has the statistical analysis been performed appropriately and rigorously?

Reviewer #1: Yes

Reviewer #2: Yes

Reviewer #3: Yes

Reviewer #4: Yes

Reviewer #5: I don't know

Reviewer #6: No

4. Have the authors made all data underlying the findings in their manuscript fully available (please refer to the Data Availability Statement at the start of the manuscript PDF file)?

Reviewer #1: Yes

Reviewer #2: Yes

Reviewer #3: Yes

Reviewer #4: Yes

Reviewer #5: Yes

Reviewer #6: No

5. Is the manuscript presented in an intelligible fashion and written in standard English?

Reviewer #1: Yes

Reviewer #2: Yes

Reviewer #3: Yes

Reviewer #4: Yes

Reviewer #5: Yes

Reviewer #6: No

Reviewer #1: The authors have addressed the feedback provided. Congratulations to the team!

Reviewer #2: (No Response)

Reviewer #3: Thank you for addressing all the comments

Reviewer #4: All three round of ZDHS are somewhat old and do not exactly reflect the current situation in Zimbabwe. This is deem a weakness and should be reflected in the discussion and limitation. The paper has improved greatly and I have no further comments to make.

Congratulations to the authors.

Reviewer #5: Dear Authors

Please refer to my comments below:

1. Line 54 - please state the regional average and add a reference

2. Line 57-59 - if the contribution of NMR to U5MR is increasing, is the UFMR still declining?

3. Line 69 - please spell out ML on the first instance here

4. Line 119 - no of ANC visits?

5. Line 119 - what timepoint was BP and urine testing done? Is this based on maternal recall or medical records?

6. Line 143 - Was imputation attempted to address missingness?

7. Why werent F1scores reported considering the imbalanced dataset?

8. Line 217 - Not sure what impact a urine sample being obtained has on neonatal mortality?

9. Line 219 - Would be good to show how many were LBW

10. Table 3 - what does wanted no more indicate?

11. Why was birth interval > 48 months? The usual practice is 24 months

12. Considering the predictive factors identified through the ML process, I dont see what additional evidence has been generated through this work.

Reviewer #6: Based on a thorough scientific review of the manuscript titled "Understanding drivers of neonatal mortality in Zimbabwe: A machine learning approach using survey data", here are detailed and scientifically grounded comments supporting rejection of the paper in its current form:

1.The machine learning models perform poorly (AUC ≤ 0.72; precision & F1 < 0.05). Such low performance limits practical utility and undermines the model's predictive value.

2.The extreme class imbalance (74:1) was managed only by down sampling, leading to potential information loss. Better methods (e.g., SMOTE, cost-sensitive learning) were not used or discussed.

3. The study adds little beyond traditional methods. Predictors identified by ML are mostly known and show minimal novelty. Variable importance was misinterpreted as causal, without robust validation.

4. No external or temporal validation was conducted. Results are based solely on internal cross-validation, limiting generalizability.

5. Important variables were dropped due to missingness. No imputation or sensitivity analysis was performed. Confounding and bias remain unaddressed.

5. The manuscript makes causal policy recommendations from purely associative, cross-sectional data without using causal inference methods.

**Do you want your identity to be public for this peer review?** For information about this choice, including consent withdrawal, please see our Privacy Policy

Reviewer #1: No

Reviewer #2: **Yes:** Edmund Mintah Wiafe

Reviewer #3: **Yes:** Dr. Omona Kizito

Reviewer #4: **Yes:** John Azaare, PhD

Reviewer #5: No

Reviewer #6: No

---

## [Editor Report · Decision Letter 2]

22 Oct 2025

PGPH-D-24-02248R2

Understanding drivers of neonatal mortality in Zimbabwe: A machine learning approach using survey data.

Dear Dr. Mbinda,

Thank you for submitting your manuscript to PLOS Global Public Health. After careful consideration, we feel that it has merit but does not fully meet PLOS Global Public Health’s publication criteria as it currently stands. Therefore, we invite you to submit a revised version of the manuscript that addresses the points raised during the review process.

EDITOR:

**Major Comments**

While *PLOS Global Public Health* accepts manuscripts of any length, a 8,000-word paper is excessive. The authors are encouraged to present and discuss their findings more concisely.Previous reviewers noted that it was unclear what added value the machine learning (ML) approach brings beyond traditional regression. The authors still have not adequately addressed why ML was necessary, how it performed better, and what insights it uncovered that classical methods may have missed. For example, including a short paragraph summarizing existing studies using ML in LMICs to better frame your contribution.The authors mention using unweighted cases (n = 16,941) for modeling and applying survey weights for frequencies only. Why were weights not used in model training or performance evaluation? If the models were trained on unweighted data, statements such as “The results from this study are representative of drivers of neonatal mortality in Zimbabwe” are misleading.The authors state that a 70/30 random split was used between training and test sets and that 10-fold cross-validation (repeated three times) was implemented during training. It is unclear whether the split or the cross-validation folds were stratified by neonatal death outcome.Please clarify whether hyperparameter tuning occurred within an inner cross-validation loop (nested CV). If not, justify the current approach or consider re-running the models using nested CV.The authors state that a total of 30 out of the 33 variables included in the model-building process were selected by the XGBoost model as relatively important, but do not explain which importance metric was used (gain, cover, or split frequency) or how correlated predictors were handled.It is unclear why Boruta was used if XGBoost was the main model. Also for the Boruta algorithm, specify whether it was applied before or after the train-test split. The authors note that Boruta was used to remove “unimportant variables.” For transparency, please report the number of confirmed, tentative, and rejected features at the end of the Boruta run.While the language has improved to reduce causal interpretation, several statements still imply causality. Additionally, throughout the manuscript, terms such as drivers, determinants, and key predictors are used interchangeably. “Drivers” and “determinants” imply causality and should be replaced with “predictors” or “associations.”Authors state “Receiving newborn postnatal care was found to be protective in this study, and this underscores the need for mothers and their babies to access and utilize appropriate health services soon after delivery (22).” Deaths occurring before postnatal visits may artificially inflate the apparent protective effect of PNC. Because PNC often occurs days after birth, only surviving neonates are “eligible,” leading to selection bias.“Lower birth weight (aOR [95% CI]: 0.9997 [0.9995–0.9999]) was positively associated with odds of neonatal mortality… interventions should focus on preventing and addressing the effect of birth weight on child survival.”  Birth weight is a strong predictor but not necessarily a modifiable causal factor.

**Minor Comments**

The aOR for birth weight (0.9997) is not intuitive. Please present it on an interpretable scale e.g. per 100 gramsGiven that several variables had >20% missing data, justify the assumption that data were missing at random (MAR).It is unclear why all the listed models were included. For instance, why include both GBM and XGBoost, given their similar architecture?There is no mention of random seed setting, which raises concerns about reproducibility not knowing if the code will be shared.Given only 473 neonatal deaths, there is a risk of overfitting, consider reporting the events-per-variable (EPV) and discuss overfitting risk.The description of the hybrid SMOTE procedure is confusing. Creating five synthetic minority samples per case and then downsampling the majority class to equal size may produce an artificially balanced dataset that departs from real-world prevalence. Potential ecological validity issues and model calibration (e.g., Platt scaling or probability adjustment) afterward may be needed.Given the high class imbalance, the authors should consider using Precision–Recall (PR) curves and AUC-PR, which may be more informative than ROC curves.The authors report precision and F1-scores around 15%, implying a high false-positive rate. This should be discussed explicitly in the Limitations section.There was no explicit discussion of multicollinearity. Please report correlation checks or variance inflation factors (VIFs).“Support Vector Mechanism (SVM)” should read “Support Vector Machine (SVM),” and “hot encoding” should be “one-hot encoding.”Early breastfeeding and newborn PNC appear protective but may be susceptible to survivor bias and reverse causation. This should be discussed in the Limitations section, and ideally, sensitivity analyses excluding early neonatal deaths or stratified by survey round should be conducted.The Limitations section does not discuss potential temporal confounding from pooling data across survey years. Did you adjust for survey year or conduct any sensitivity analyses by survey round?In the Discussion, the authors highlight many variables as “relatively important” in XGBoost but not significant in logistic regression. Please explain why this might occur (e.g., nonlinear effects, interactions, or differing sensitivity to correlated predictors).In the discussion of the strengths and limitations of XGBoost, include comments on calibration, and external validation.For the imputation process, specify the missRanger settings, number of imputations (single vs. multiple), whether the imputation included the outcome variable, and how imputation uncertainty was handled or propagated.

A rebuttal letter that responds to each point raised by the editor. You should upload this letter as a separate file labeled 'Response to Reviewers'.A marked-up copy of your manuscript that highlights changes made to the original version. You should upload this as a separate file labeled 'Revised Manuscript with Track Changes'.An unmarked version of your revised paper without tracked changes. You should upload this as a separate file labeled 'Manuscript'.

We look forward to receiving your revised manuscript.

Kind regards,

Mufaro Kanyangarara

Academic Editor

---

## [Editor Report · Decision Letter 3]

5 Jan 2026

Understanding drivers of neonatal mortality in Zimbabwe: A machine learning approach using survey data.

PGPH-D-24-02248R3

Dear Mr. Mbinda,

We are pleased to inform you that your manuscript 'Understanding drivers of neonatal mortality in Zimbabwe: A machine learning approach using survey data.' has been provisionally accepted for publication in PLOS Global Public Health.

Best regards,

Mufaro Kanyangarara

Academic Editor
